# *Laguncularia racemosa* Phenolics Profiling by Three-Phase Solvent System Step-Gradient Using High-Performance Countercurrent Chromatography with Off-Line Electrospray Mass-Spectrometry Detection

**DOI:** 10.3390/molecules26082284

**Published:** 2021-04-15

**Authors:** Fernanda das Neves Costa, Gerold Jerz, Peter Hewitson, Fabiana de Souza Figueiredo, Svetlana Ignatova

**Affiliations:** 1Institute of Natural Products Research, Federal University of Rio de Janeiro, CCS, Bloco H, Ilha do Fundão, RJ 21941-11 590, Brazil; fabiana.farmacia@gmail.com; 2Institute of Food Chemistry, Technische Universität Braunschweig, Schleinitzstrasse 20, 38106 Braunschweig, Germany; g.jerz@tu-braunschweig.de; 3Advanced, Bioprocessing Centre, Department of Chemical Engineering, Brunel University London, London UB8 3PH, UK; peter.hewitson@brunel.ac.uk (P.H.); svetlana.ignatova@brunel.ac.uk (S.I.)

**Keywords:** high-performance countercurrent chromatography, off-line MS/MS detection, three-phase solvent system, step-gradient, *Laguncularia racemosa*

## Abstract

The detailed metabolite profiling of *Laguncularia racemosa* was accomplished by high-performance countercurrent chromatography (HPCCC) using the three-phase system *n*-hexane–*tert*-butyl methyl ether–acetonitrile–water 2:3:3:2 (*v*/*v*/*v*/*v*) in step-gradient elution mode. The gradient elution was adjusted to the chemical complexity of the *L. racemosa* ethyl acetate partition and strongly improved the polarity range of chromatography. The three-phase solvent system was chosen for the gradient to avoid equilibrium problems when changing mobile phase compositions encountered between the gradient steps. The tentative recognition of metabolites including the identification of novel ones was possible due to the off-line injection of fractions to electrospray ionization mass spectrometry (ESI-MS/MS) in the sequence of recovery. The off-line hyphenation profiling experiment of HPCCC and ESI-MS projected the preparative elution by selected single ion traces in the negative ionization mode. Co-elution effects were monitored and MS/MS fragmentation data of more than 100 substances were used for structural characterization and identification. The metabolite profile in the *L. racemosa* extract comprised flavonoids, hydrolysable tannins, condensed tannins and low molecular weight polyphenols.

## 1. Introduction

Countercurrent chromatography (CCC) is an all-liquid method, with no solid support, in which the stationary liquid phase is retained in the apparatus using centrifugal force only [1]. The principle behind this technique underlies the partitioning of a sample in a biphasic liquid solvent system [2]. Among many advantages, the technique is highly versatile; has high loading capacity; is easy to scale up; and eliminates sample loss by chemical degradation and irreversible adsorption [1,3,4]. CCC is a powerful tool in the phytochemical working field as it enables plant extract fractionation with existing major compounds and also isolation or fortification of minor compounds [5,6]. This characteristic is even more noticeable when semi-preparative and preparative scales are employed [7,8]. Standard CCC separation experiments do not provide the full flexibility, solely operating with a two-phase solvent system, and the isocratic elution–extrusion mode [9,10].

To improve the polarity range in the CCC operation field, three-phase solvent systems were developed/applied, due to the great differences in polarities between the upper, middle and lower phases. Recently, some applications have been reported but only half of them actually were used as three phases in the separation process [11,12,13,14,15]. Tri-phasic systems are built from a two-phase system normally composed of n-hexane, acetonitrile and water in combination with a fourth solvent such as methyl acetate, ethyl acetate, methyl *tert*-butyl-methyl-ether or dichloromethane to create the third phase. Very few tri-phasic systems are described due to limited solvent combinations that form these three stable phases in a convenient volume percentage [11].

In analogy to solid phase chromatography, gradient elution in CCC intends to shorten the duration of a separation process and may also improve resolution. A common way to perform gradient elution is to change the mobile phase polarity over time [16], although gradient elution mode in CCC is less frequently used as the biphasic liquid system is in the equilibrium state and the change of composition in one phase corresponds directly to a change in the respective other liquid phase [17]. However, if a three-phase solvent system is used for gradient elution purposes, all phases involved for the experiment were previously in contact. Disturbance of the equilibrium and collapse of phase layers are omitted during the separation process while maintaining the broad polarity range for the recovery process.

In this work, a three-phase solvent system in step-gradient elution mode high-performance countercurrent chromatography (HPCCC) with off-line ESI-MS/MS detection was used for metabolite profiling of the ethyl acetate partition of the leaves from the mangrove plant *Laguncularia racemosa*. A similar approach was used on *Anogeissus leiocarpus* for compound identification but using centrifugal partition extraction (CPE), and off-line NMR detection [15]. *L. racemosa* (Combretaceae), popularly known as white mangrove, is the only occurring specie in the genus [18], and is considered as a strict mangrove [19], characteristic for growing in brackish coastal environments [20], and with excellent function for stabilizing shorelines against erosion [21]. From aspects of ethno-medicinal use, the plant is applied as astringent and tonic for dysentery and fever [22]. To date, there are only a few studies on the production of secondary metabolites [23,24,25,26,27], probably due to its high complexity of polar natural products. 

## 2. Results and Discussion

The composition of *L. racemosa* ethyl acetate solvent partition (EtOAcPart) was initially investigated by TLC and LC-ESI/TOFMS analysis (Appendix A). The liquid mass-spectrometry profile showed a high chemical complexity containing metabolites in a larger polarity range, making this mixture an ideal case study for the application of three-phase gradient elution. Some multiple-solvent biphasic systems have been proposed to great extend the polarity in CCC [28]. However, as the phases are in steady mixing contact during the complete separation process, a change of the mobile phase composition during gradient elution directly influences the stationary phase composition as well, and as consequence, disturbs the equilibrium and could lower or even lead to low chromatographic resolution [29,30]. To circumvent the equilibrium obstacle during the gradient elution procedure, a three-phase solvent system was used instead of two (or more) different biphasic systems. In this approach all phases involved in the separation were previously saturated with each other. The tri-phasic system *n*-hexane–*tert*-butyl-methyl ether–acetonitrile–water 2:3:3:2 (*v*/*v*/*v*/*v*) was used in the semi-preparative purification of *L. racemosa* EtOAcPart.

### 2.1. HPCCC of L. racemosa Metabolites by Off-Line ESI-MS/MS Profile Detection in the Sequential Order of Recovery

The *L. racemosa* EtOAcPart was separated by semi-preparative HPCCC chromatography, and the off-line injection profiling by injections of recovered fractions to ESI-MS/MS distinguished 17 principal phenolic constituents (Figure 1). However, as a result of the highly concentrated injections of respective HPCCC fractions, the selected single ion-based projection of the HPCCC experiment revealed more than 100 different metabolites (**1**–**109**) (Partially shown in Figure 1 and Appendix A). Preliminary LC-ESI/TOF MS analysis was not capable of detecting all minor compounds due to concentration levels below the detection limits.

Large advantage of the off-line injection profiling methodology of preparative HPCCC-fractions by an ESI-MS detector in the sequence of recovery is the ‘on-the-fly’ delivery of the respective molecular weight- and MS/MS-fragmentation data of all ionizeable compounds in one step. This is a very fast process to get the required data for immediate compound identifications in the respective HPCCC-fractions. A full mass-spectrometry guided metabolite profile with more than ten automatically selected precuresur ions for MS/MS on a larger lab-scale preparative HPCCC fractionation can be achieved in a 60 to 100 min experimental mass-spectrometry time frame. This mass-spectromtry approach is roughly by a factor of hundred faster than the single investigation of resepctive HPCCCC fractions by LC-ESI-MS/MS analysis. The results displayed in a single data file are ready to use and not mutiple analysis sets need to be compared for guiding the decisions in fractionation process. This powerful approach was previously applied by Costa et al. [8] on the complex metabolite mixture extracted from the Brazilian plant *Salicornia gaudichiana*.

In case of the investigated *L. racemosa* ethylacetate solvent partition, the elution ranges of a large selection of higher and lower concentrated target molecules (Table 1) were visualized in the recovered HPCCC-fractions by selected single ion traces for performing the accurate fractionation, recovery and preventing unintentional mixing of already separated compounds. Additionally, the existing compound co-elution effects, and the sequential elution orders of separated isobars/ isomers were clearly detected and visualized (Figure 1).

One of the special cases of isomer/isobar separation by HPCCC is the selected ion trace [M − H]^−^ at *m*/*z* 615, as the HPCCC experiment separated flavonoid-glycosides with identical molecular weights as displayed in the low resolution ESI-MS injection profile (Figure 1). A set of two partly co-eluting positional isomers of myricetin-desoxy-hexoside-gallate (**37**) (fraction range 91–115) were absolutely separated from the later eluting isobar quercetin-hexoside-gallate (**38**) (fraction range 137–153). 

The selected ion trace at *m*/*z* 635 displayed two strong HPCCC elution ranges with compeletly separated compound areas with **41** (range 45–57), and **77** (133–151) (Figure 1). However, the metabolite **41** with lower elution volume in the HPCCC run was identified by the ESI-MS/MS profile data as myricetin whereby the ESI-ion-source dimer [2M − H]^−^ was generated in dominant intensity. This was confirmed by the exact identical position of *m*/*z* 317 ([M − H]^−^) in the HPCCC profile. Nevertheless, the late eluting metabolite **77** was identified as a [M − H]^−^-signal with a hexosid unit substituted by three galloyl-moieties indicated by MS/MS neutral loss cleavage (Δ*m*/*z* 152) to *m*/*z* 483, and 331 of gallic acid releases. The tetra-galloyl-hexoside with [M – H]^−^ at *m*/*z* 787 (**82**) (Table 1) co-eluted in this HPCCC run as seen in Figure 1, as well as the penta-galloyl-hexoside (**85**) (Table 1). A very large elution volume for recovery in the triphasic HPCCC experiment displayed the galloyl diHHDP hexoside (**83**) seen by [M − H]^−^ at *m*/*z* 935 (range 145–167) (Figure 1 and Table 1). Although the constitution of certain compounds had been different, the polarity differences were not sufficient for a successful HPCCC separation as seen for the selected ion traces [M – H]^−^
*m*/*z* 585 (quercetin pentoside gallate, **32**), and *m*/*z* 599 (quercetin desoxyhexoside gallate, **33**) (Figure 1).

Using literature to guide the identification process of the minor, and very minor concentrated derivatives, literature was verified and the few previously isolated compounds in *L. racemosa* were listed with molecular weights as a comparative database. Most of the unknown compounds were characterized by ESI-MS/MS fragmentation and indicative neutral loss pattern. High accuracy molecular weights acquired by LC-ESI/TOF MS were used to ratify and/or verify the proposed molecular formulas. Phytochemical investigations describing chemical compounds on other genus of Combretaceae helped to support the results based on chemotaxonomic knowledge. From the aspect of natural product classes, the chemical composition of the EtOAcPart was distinguished in four main groups as flavonoids, hydrolysable tannins, condensed tannins and other low molecular weight polyphenols (Appendix A). The chemical structures and substitution patterns of fractionated and identified compounds are shown in Figure 2.

Fractions had been combined on the basis of TLC analysis and the electrospray mass-spectrometry profiling experiment. Appendix A displays the TLC-analysis on the combined fractions of the HPCCC experiment. Table 1 lists HPCCC chromatographic elution, and ESI-MS/MS informations; LC ESI-TOF-MS data (when present) and tentative identification. Although *L. racemosa* EtOAcPart showed quite complex constituents, most phenolic compounds were well separated.

#### 2.1.1. Flavonoids and Derivatives

Flavonoid derivatives were detected and identified in *L. racemosa* EtOAcPart by ESI-MS/MS as principal compounds in the recovered HPCCC fractions (Table 1, compounds **1-56**). Flavonoids including flavonols, flavones, flavanols and flavanones were found in free form, linked to one sugar unit, as well as in the presence of galloyl substituents. The tentative identification of the flavonoid-aglyca (compounds **1**–**13**) was done by comparison to specific fragmentation patterns, as the spectra of this flavonoid often displayed loss of small neutral fragments contributing to structure information [31,32,33]. The free flavonoid aglyca eluted during the first step of the gradient before the glycoside linked flavonoids, in accordance to mobile phase/compound polarity in the tail-to-head mode. 

The flavonoid-*O*-substituted characteristically exhibited the neutral loss [34] attributed to a pentose unit [M − H − 132]^−^, hexose unit [M − H − 162]^−^, desoxy-hexose unit [M − H − 146]^−^, glucuronyl unit [M − H − 176]^−^, galloyl moiety [M − H − 152]^−^ and combination of these substituents. A set of quercetin-*O*-pentoside, -*O*-desoxy-hexoside, -*O*-hexoside, -*O*-glucuronide and myricetin-*O*-pentoside, -*O*-desoxy-hexoside, -*O*-hexoside were detected in compounds **18**, **21**, **22**, **25**, **26**, **30** and **31** [35,36,37,38,39,40,41]. The substituent gallate was found connected to (epi)-catechin (20), (epi)-gallocatechin (23) and myricetin (28) as well as in glycosylated forms of quercetin and myricetin (**32**–**34**, **37**–**40**) [41]. The digallate derivative of quercetin and myricetin-*O*-pentoside were also recognized in compounds **45** and **46**.

Aglycones apigenin, kaempferol, quercetin and tricin were previously reported in *L. racemosa* [8,26] in addition to the glycosylated derivatives quercetin-3-*O*-arabinoside and quercetin-3-*O*-rhamnoside [24]. Not fully identified derivatives could be distinguished by observed aglycone fragment ions in MS/MS.

#### 2.1.2. Hydrolysable Tannins

Hydrolysable tannins, well-known in Combretaceae, were the second main class of natural compounds detected by the HPCCC and off-line injection ESI-MS/MS experiment (Table 1, compounds **57**–**86**) [36,37,38,39,40,42,43]. It included derivatives of gallic acid, ellagic acid, gallotannins and ellagitannins. Some of the ellagic acid and its methyl-, dimethyl- and trimethyl ether derivatives were previously reported in *L. racemosa* [26]. Several studies describing the detection of hydrolysable tannins in species of Combretaceae can be found [44,45,46].

Common neutral loss cleavages observed in the MS/MS for simple gallic acid and its derivatives were related to the cleavage of carboxyl [M − H − 44]^−^, methyl [M − H − 15]^−^, ethyl [M − H − 29]^−^ and galloyl [M − H − 152]^−^. They were found as ester or ether arrangements. Compounds **57**–**59**, **62**, **63**, **66** and **68** were identified as gallic acid, methyl gallate, ethyl gallate, galloyl gallate, galloyl shikimate, galloyl methyl gallate and galloyl ethyl gallate, respectively [37,38,42]. 

Ellagic acid derivatives were characterized by the fragment ion *m*/*z* 301. At this point, LC ESI-TOF-MS was essential to distinguish derivatives from quercetin and ellagic acid. The sequence of compounds comprised ellagic acid itself and the -methyl, -dimethyl, -trimethyl, -pyrogalloyl and dihexoside ether forms (**60**, **61**, **64**, **67**, **69**, **74**) [36,39,40,43]. Additionally, valoneic acid dilactone (**70**) and its ethyl ether derivative (**73**) were detected [36,40]. 

By comparison to literature [47], the molecular masses of compounds **65**, **72**, **77**, **82** and **85** showed that they consist of a gallotannin series of molecules (mono-, di-, tri-, tetra- and penta-galloyl hexosides) [37,38,42]. A similar series of monomeric ellagitannins (HHDP-, NHDP-, HHDP galloyl-, diHHDP-, HHDP digalloyl-, diHHDP galloyl- and HHDP trigalloyl-) were found in compounds **71**, **75**, **76**, **80**, **81**, **83** and **84** [36,38,40]. The ellagic acid punicalin (**79**) was further detected at *m*/*z* 781.

#### 2.1.3. Condensed Tannins

Condensed tannins (proanthocyanidins), formerly observed in *L. racemosa* wood and leaves [48,49,50], were recognized and characterized based on the detected flavanol-aglyca (**4**, **7**, **9**) and its gallate derivatives (**20**, **23**). They were found as homo-dimers consisting of (epi)-catechin (**87**), (epi)-gallocatechin (**89**) and (epi)-gallocatechin gallate (**91**) [40]. Additionally, as hetero-dimers, existing as (epi)-catechin-(epi)-gallocatechin (**88**) and (epi)-catechin gallate-(epi)-gallocatechin gallate (**90**). The trimeric (epi)-gallocatechin (**92**) was also encountered. Compounds had fragmentation patterns related to the cleavage of flavanol units according to literature [51]: [M − H − 289]^−^ for (epi)-catechin loss, [M − H − 305]^−^ (epi)-gallocatechin loss, [M − H − 441]^−^ (epi)-gallocatechin gallate loss and [M − H − 162]^−^ for gallate loss.

Considering the elution order of compounds in respect to gradient polarity range, the flavonol-aglyca eluted before the gallate derivatives, both in the first step, while dimers and trimers stayed retained in the column until extrusion started.

#### 2.1.4. Low Molecular Weight Polyphenols

Other compounds were recognized and characterized based on precursors/derivatives of existing identified compounds in the off-line ESI-MS/MS profile or on the *L. racemosa* chemical database. Simple phenolic compounds included catechol (**93**) and pyrogallol (**95**), common occurring products in the hydrolysable tannins pathway [35]. Benzoic acid derivatives with frequent [M − H − 44]^−^ corresponding to the neutral loss of CO_2_, comprised protocatechuic (**96**) and vanillic (**97**) acids [37]. The amino derivatives aminocatechol (**94**) and amino protocatechuic acid (**98**) were also detected [52]. The chromone detected at *m*/*z* 193, was identified as trihydroxy-chromone (**99**) and had its molecular formula confirmed by HRMS.

The jasmonic acid (**100**) and its sulphated derivative 5′-hydroxy-sulphonyloxy jasmonic acid (**105**), earlier isolated from the *L. racemosa* twigs and leaves [23], could be found at *m*/*z* 209 and 305, respectively. ESI/TOF MS data confirmed the proposed compounds. Ordinary oleic and linoleic fatty acids (**102-104**), jasmonic acid biosynthetic precursor, were further encountered. Another sulphated derivative isolated from *L. racemosa* leaves [26] was found at [M – H]^−^ at *m*/*z* 707 and was identified as integracin D (**107**) [26]. Due to concentration limits, the compound could not be detected in the ESI/TOF MS analysis and structure was not fully confirmed.

## 3. Materials and Methods

### 3.1. Chemical Reagents and Solvents

Preparation of extracts was carried out with analytical grade solvents from Tedia Brazil (Rio de Janeiro, Brazil). LC-ESI/TOF-MS/MS analyses used HPLC grade solvents from Tedia Brazil (Rio de Janeiro, Brazil). HPCCC separations were performed with analytical grade solvents from Fisher Chemicals (Loughborough, UK). ESI-MS/MS analyses were done with HPLC grade solvents from VWR Chemicals (Radnor, PA, USA). NMR analyses used deuterated solvents from Cambridge Isotope Laboratories (Tweksbury, MA, USA) and TMS as internal standard. All aqueous solutions were prepared with pure water produced by Milli-Q water (18.2 MΩ) system (Thame, UK).

### 3.2. Preparation of the Extract

*Laguncularia racemosa* (3 kg) was collected at Guaratiba Biological and Anthropological Reserve (Rio de Janeiro, Brazil) in November 2010. Specialist researchers from the Nucleus of Mangrove Studies (University of the State of Rio de Janeiro) helped in the localization, identification and collection of the plant. The leaves were dried and grounded in a laboratory mill (Laboratory Retsch mill, Haan, Germany) and 1800 g were submitted to maceration with ethanol–water 8:2 (*v*/*v*) in 10 cycles of 24 h. The solvent was evaporated under reduced pressure at 50 °C and the crude extract (255 g) was partitioned between water and organic solvents, affording different extracts: n-hexane (4 g), dichloromethane (8 g), ethyl acetate (15 g) and aqueous (215 g).

### 3.3. Thin Layer Chromatography

Preliminary analyses of EtOAcPart, solvent system evaluation tests and CCC fraction analyses were done by thin layer chromatography (TLC) on normal phase silica gel TLC plates (SiO_2_-60, F254, Merck, Darmstadt, Germany, gel 60 RP-18, F254S) developed with EtOAc–acetone–H_2_O 25:15:10 (*v*/*v*/*v*), and acetonitrile-H_2_O 1:1 (*v*/*v*) for reversed phase C18-plates (RP18W, Macherey and Nagel, Düren, Germany). Results were visualized by using spray-reagent H_2_SO_4_ (10% *m*/*v*) in methanol with vanillin 5% in ethanol and flash heating on a hot plate 105 °C.

### 3.4. LC-ESI/TOF MS Preliminary Analysis

The EtOAcPart was also analysed by LC–ESI/TOF-MS with a 1200 Series LC-chromatograph (Agilent, Palo Alto, CA, USA) coupled with a MicrOTOF II time-of-flight mass spectrometer (Bruker Daltonics, Inc., Billerica, MA, USA). 5 µL injection was performed with an autosampler on a Poroshell EC-C18 column (100 × 2.1 mm; 2.7 µm, Agilent, Palo Alto, CA, USA). The source temperature was set at 200 °C, the drying gas (nitrogen) flow rate was 10.0 L/min and the nebulizer gas (nitrogen) pressure was 4 bar. Data were acquired in negative mode in the range of *m*/*z* 100–1500. The capillary voltage was 3.8 kV, the capillary exit voltage was −150 V, the skimmer 1 and 2 voltages were 50 V and 23 V, respectively, the hexapole 1 voltage was set to −23 V, the hexapole RF voltage was 120 Vpp, lens 1 transfer was 68 μs and lens 1 pre plus stage was 7 μs. Mass calibration was achieved by infusing ammonium formate in an isopropanol–water mixture (1:1, *v*/*v*) as an external standard. All data were analysed using Bruker Daltonics ESI Compass Data Analysis Version 4.0 SP 1 (Bruker Daltonics Inc., Billerica, MA, USA). The mobile phase consisted of spectroscopic grade methanol (B) and ultrapure water (A) containing 0.05% (*v*/*v*) formic acid. The linear gradient elution was set from 10% to 100% of B in 90 min at a flow rate of 0.3 mL/min.

### 3.5. High Performance Countercurrent Chromatography

#### 3.5.1. Equipment

CCC separations were performed on a semi-preparative HPCCC system (model Spectrum, Dynamic Extractions Ltd., Gwent, UK) equipped with two counter-balanced bobbins with perfluoroalkoxypolymer (PFA) tubing (1.6 mm i.d.) wound in multi-layer coiled-columns, resulting in 143.5 mL total volume (V_C_). The rotation speed was adjusted to the maximum velocity of 1600 rpm (240 g). Solvent phase systems were delivered by a constant flow pump (Agilent HP1200, Palo Alto, CA, USA) to the HPCCC system. A semi-preparative sample loop (7.15 mL) was used to inject the dissolved sample over a low-pressure valve (Upchurch Model V-450, with 1.6 mm i.d. fittings) to the chromatographic system. Fractions were collected by a fraction collector (Agilent HP1200, Palo Alto, CA, USA).

#### 3.5.2. Three-Phase Solvent System Test Evaluation

The three-phase solvent systems were composed of *n*-hexane–methyl acetate –acetonitrile–water and *n*-hexane–*tert*-butyl methyl ether–acetonitrile–water [11,53,54]. For the experiments for solvent system evaluation, 2 mg of the EtOAcPart were dissolved in a test tube containing 2 mL of each phase of the thoroughly equilibrated solvent systems. The test tubes were shaken vigorously for compound partition. After the phase layers had completely separated and distribution equilibrium was established, the resulting phase layers were analyzed by TLC (Appendix A).

#### 3.5.3. Solvent System and Sample Preparation

The selected solvent system *n*-hexane–*tert*-butyl methyl ether–acetonitrile–water (2:3:3:2, *v*/*v*/*v*/*v*) was thoroughly equilibrated in a separatory funnel at room temperature. The three phases were separated shortly before use and degassed by ultra-sonication for 5 min. The sample solution was prepared by dissolving the sample at fixed concentration (100 mg/mL) and coil-volume (5% V_C_) in the lower aqueous phase only.

#### 3.5.4. HPCCC Separation Procedure

Separation was performed in a normal step-gradient elution mode. The more aqueous lower phase was used as the stationary phase while organic upper and middle phases were used as mobile phases as shown in Figure 3. The system was completely filled with the lower aqueous stationary phase. Rotation was set to 1600 rpm. For the separation, the upper organic mobile phase was pumped at 4.0 mL/min. After reaching hydrodynamic equilibrium, the sample was injected to the HPCCC column. For the first elution step, 214.5 mL mobile phase (1.5 V_C_) of upper phase was pumped through. For the second elution step, 2 V_C_ (286 mL) of the middle phase was pumped through the HPCCC system. Fractions were collected at 1 min intervals. For the extrusion step, rotation was reduced to 200 rpm and the column contents were pushed out of the system by lower phase at 8.0 mL/min and fractions were collected at 30 s intervals. The temperature control was maintained at 30 °C.

### 3.6. Metabolite Profiling by Offline Injections to ESI-MS/MS

The molecular weight profiles of the recovered HPCCC fractions were monitored by off-line ESI-MS/MS and were recorded in a single data file using the ion-trap mass-spectrometer HCT-Ultra ETD II (Bruker Daltonics, Bremen, Germany). Aliquots of 0.75 μL of odd numbered CCC fractions were directly filled to vials, dried and redissolved in 1.0 mL of methanol for conducting the ESI-MS analysis. Fractions were delivered to the ESI-MS/MS by a HPLC-pump (binary pump, G1312 A, 1100 Series, Agilent, Waldbronn, Germany) using the make-up solvent system with a flow rate of 0.5 mL/min composed of acetonitrile and water (1:1, *v*/*v*). ESI-MS/MS parameter settings were in the negative ionization mode, with scan-range between *m*/*z* 100–2000, where mostly deprotonated [M – H]^−^ ion signals were generated. An auto-MS/MS method fragmented the nine most intense peaks and to monitor and characterize co-eluting compounds. Drying gas was nitrogen (flow rate 10.0 L/min, 310 °C), and nebulizer pressure was set to 60 psi. Ionization voltage at HV capillary was 3500 V, HV end plate off set −500 V, trap drive 61.8, octupole RF amplitude 187.1 Vpp, lens 2 60.0 V, Cap Ex −115.0 V, max. accumulation time 200 ms, averages 5 spectra, trap drive level 120%, target mass range: *m*/*z* 500, compound stability 80%, Smart ICC target 70.000, ICC charge control ‘on’ and smart parameter setting ‘active’.

## 4. Conclusions

The combination of analysis of preparative HPCCC fractions with off-line injections to an ESI-MS/MS device was proven to be highly effective for a full metabolite chemical profile for polyphenols using the negative ionization mode. The use of a three-phase solvent system for HPCCC in a step-gradient elution mode was adequate to maintain equilibrium and chromatographic resolution while improving mobile phase strength. The ESI-MS/MS projection of the semi-preparative HPCCC experiment visualized over 100 compounds by selected single ion traces and was an adequate confirmation of the LC–ESI/TOF MS analysis. This study detected a variety of metabolites from different classes occurring in *L. racemosa* EtOAcPart and used chemotaxonomic data to guide the MS/MS putative structure elucidation.

## Figures and Tables

**Figure 1 molecules-26-02284-f001:**
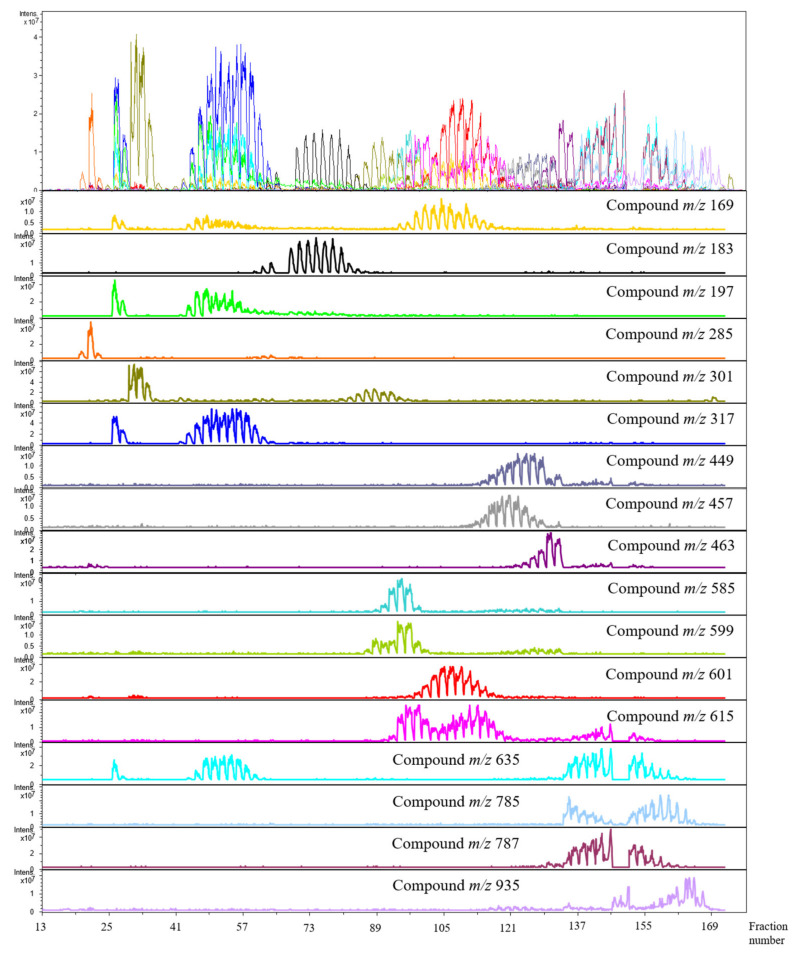
Selected electrospray ionization mass spectrometry electrospray ionization mass spectrometry ions traces (negative mode) of phenolics of *L. racemosa* EtOAcPart detected in the off-line injected high-performance countercurrent chromatography (HPCCC) fractions. HPCCC separation using *n*-hexane- *tert*-butyl-methyl ether–acetonitrile-water 2:3:3:2 (*v*/*v*/*v*/*v*) as triphasic solvent system.

**Figure 2 molecules-26-02284-f002:**
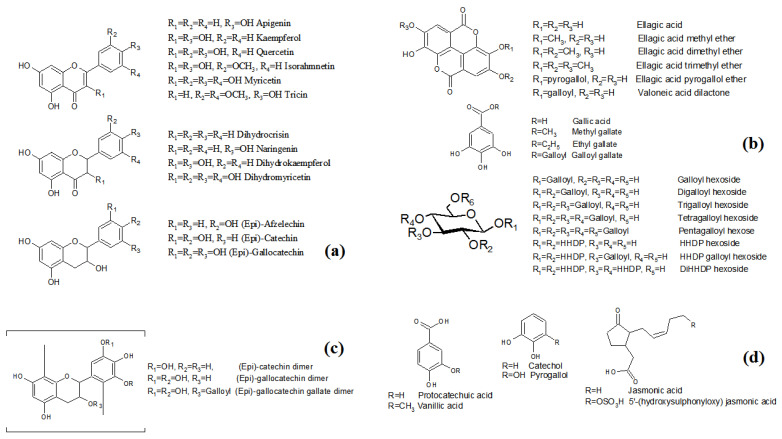
*Laguncularia racemosa* EtOAcPart general structures and tentative substitution patterns of some of the existing compounds. (**a**) Flavonoids, (**b**) hydrolysable tannins, (**c**) condensed tannins and (**d**) other low molecular weight polyphenols.

**Figure 3 molecules-26-02284-f003:**
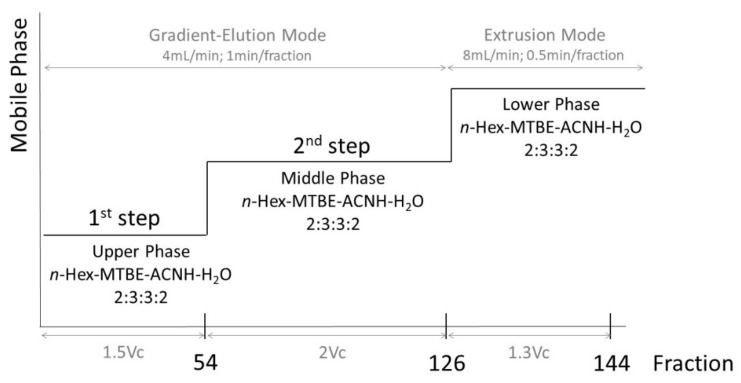
HPCCC three phase solvent system step-gradient procedure.

**Table 1 molecules-26-02284-t001:** Detected compounds in the HPCCC off-line ESI-MS/MS phenolic profile of *Laguncularia racemosa* EtOAcPart.

Cpd	CCC-Fraction	MS [M – H]−(*m*/*z*) MS/MS [M – H]− (*m*/*z*)	LC-RT(min)	ESI/TOF MS Formula (Error in ppm)	Identification
Flavonoids and derivatives					
**1**	11–15	255 237, 226, **209**, 156	n.d.	-	Dihydrocrysin
**2**	21	269 151	13.7	269.04593 C_15_H_9_O_5_ (1.4)	Apigenin
**3**	19–21	271 177, 151	1.9	271.04885 C_15_H_11_O_5_ (45.6) *	Naringenin
**4**	23	273 167	29.1	273.08108 C_15_H_13_O_5_ (15.5)	Afzelechin
**5**	19	285 **257**, 151	40.8	285.04413 C_15_H_9_O_6_ (12.6)	Kaempferol
**6**	31–41	287 **259**, 151	12.1	287.05883 C_15_H_11_O_6_ (9.5)	Dihydrokaempferol
**7**	97–115	289 **245**, 205	6.2	289.07438 C_15_H_13_O_6_ (9.0)	(Epi)-catechin
**8**	29–33	301 179, **151**	35.4	301.03937 C_15_H_9_O_7_ (13.3)	Quercetin
**9**	17–19	305 **287**, 249	2.4	305.0706 C_15_H_13_O_7_ (12.8)	(Epi)-gallocatechin
**10**	21–23	315 300	42.8	315.0466 C_16_H_11_O_7_ (14.1)	Isorhamnetin
**11**	25–27	317 **179**, 151	28.9	317.03536 C_15_H_9_O_8_ (16.0)	Myricetin
**12**	81–93	319 193	9.1	319.04883 C_15_H_11_O_8_ (9.0)	Dihydromyricetin
**13**	33–41	329 **314**, 299	21.7	329.05816 C_17_H_13_O_7_ (25.9) *	Tricin
**14**	133–149	393 **317**, 241, 169	-	-	Myricetin derivative
**15**	29–33	415 301	-	-	Quercetin alkyl derivative
**16**	127–131	419 305	-	-	(Epi)-gallocatechin alkyl derivative
**17**	55–57	431 317	-	-	Myricetin alkyl derivative
**18**	97–105	433 **301**, 179, 151	27.5	433.08215 C_20_H_17_O_11_ (10.4)	Quercetin pentoside
**19**	89–91	433 319, 193	-	-	Dihydromyricetin alkyl derivative
**20**	91–95	441 289	19.3	441.08208 C_22_H_18_O_10_ (1.5)	(Epi)-catechin gallate
**21**	105–119	447 301	29.3	447.09851 C_21_H_19_O_11_ (11.7)	Quercetin desoxyhexoside
**22**	115–133	449 **317**, 316	22.4	449.07395 C_20_H_17_O_12_ (3.1)	Myricetin pentoside
**23**	113–127	457 331, 305, **169**	12.1	457.07859 C_22_H_18_O_11_ (2.1)	(Epi)-gallocatechin gallate
**24**	21–23	461 **443**, 381, 301, 193	-	-	Quercetin derivative
**25**	123–131	463 **317**, 316	24.6	463.09083 C_21_H_19_O_12_ (5.7)	Myricetin desoxyhexoside
**26**	141–145	463 301	25.9	463.09187 C_21_H_19_O_12_ (7.9)	Quercetin hexoside
**27**	153–165	467 458, **391**, 301, 169	-	-	Quercetin derivative
**28**	67–79	469 317	n.d.	-	Myricetin galatte
**29**	85–91	471 319, 301, **193**	-	-	Dihydromyricetin alkyl derivative
**30**	115–131	477 **301**, 179	15.2	477.06812 C_21_H_17_O_13_ (1.4)	Quercetin glucuronide
**31**	133–163	479 **317**, 316	22.1	479.08428 C_21_H_19_O_14_ (2.4)	Myricetin hexoside
**32**	89–97	585 433, **301**	32.7	585.09204 C_27_H_21_O_15_ (5.9)	Quercetin pentoside gallate
**33**	85–95	599 447, **301**	26.8	599.10714 C_28_H_23_O_15_ (4.8)	Quercetin desoxyhexoside gallate
**34**	95–113	601 **449**, 317	28.6	601.08787 C_27_H_21_O_16_ (7.3)	Myricetin pentoside gallate
**35**	29–33	603 301	n.d.	-	Quercetin [2M − H]^−^
**36**	125–131	611 305	-	-	(Epi)-gallocatechin [2M − H]^−^
**37**	91–115	615 463, **317**, 179	23.4	615.1014 C_28_H_23_O_16_ (3.6)	Myricetin desoxyhexoside gallate
**38**	137–153	615 **463**, 301	31.6	615.10412 C_28_H_23_O_16_ (8.1)	Quercetin hexoside gallate
**39**	113–127	629 **477**, 317, 316, 289	21.5	629.07893 C_28_H_21_O_17_ (0.8)	Quercetin glucuronide gallate
**40**	133–139	631 **479**, 317	28.9	631.09859 C_28_H_23_O_17_ (7.2)	Myricetin hexoside gallate
**41**	45–57	635 317	n.d.	-	Myricetin [2M − H]^−^
**42**	87–91	639 **319**, 301	11.2	639.05562 C_29_H_19_O_17_ (11.2)	HHDP Dihydromyricetin
**43**	25	657 317	-	-	Myricetin derivative
**44**	89	697 599	-	-	Quercetin desoxyhexoside gallate derivative
**45**	89–93	737 585, 301	n.d.	-	Quercetin pentoside digalloyl
**46**	93–103	753 601, 449, 317	32.5	753.09740 C_34_H_25_O_20_ (3.9)	Myricetin pentoside digalloyl
**47**	85–89	773 471, 301	-	-	Quercetin derivative
**48**	97–103	867 **433**, 301	n.d.	-	Quercetin pentoside [2M − H]^−^
**49**	117–127	883 **449**, 317	-	-	Myricetin pentoside derivative
**50**	115–123	892 **457**, 433	-	-	(Epi)-gallocatechin gallate derivative
**51**	117–127	899 463, **449**, 317	-	-	Myricetin pentoside derivative
**52**	87–93	901 **599**, 301	-	-	Quercetin desoxyhexoside gallate derivative
**53**	111–121	905 469, **457**, 447, 425, 301	-	-	Quercetin desoxyhexoside derivative
**54**	113–125	907 **449**, 317	-	-	Myricetin pentoside derivative
**55**	113–123	915 457	-	-	(Epi)-gallocatechin gallate [2M − H]^−^
**56**	129–131	927 **463**, 317	n.d.	-	Myricetin desoxyhexoside [2M − H]^−^
Hydrolisable tannins and deivatives					
**57**	93–115	169 125	12.1	169.01664 C_7_H_5_O_5_ (14.2)	Gallic acid
**58**	61–81	183 124	5.9	183.01418 C_8_H_7_O_5_ (12.2)	Methyl gallate
**59**	43–57	197 **169**, 125	14.3	197.04741 C_9_H_9_O_5_ (9.5)	Ethyl gallate
**60**	81–93	301 283, 257, **229**, 163	18.5	300.99939 C_14_H_5_O_8_ (1.3)	Ellagic acid
**61**	19–21	315 300	6.1	315.01809 C_15_H_7_O_8_ (10.9)	Ellagic acid methyl ether
**62**	97–103	321 169	3.9	321.03300 C_14_H_9_O_9_ (24.3) *	Galloyl gallate
**63**	133–151	325 169	3.3	325.06016 C_14_H_13_O_9_ (11.3)	Galloyl shikimate
**64**	33–39	329 314	44.2	329.02154 C_16_H_9_O_8_ (8.9)	Ellagic acid dimethyl ether
**65**	163	331 **271**, 169, 125	11.5	331.06888 C_13_H_15_O_10_ (5.5)	Galloyl hexoside
**66**	81–87	335 183	9.2	335.02817 C_15_H_11_O_9_ (37.9) *	Galloyl methyl gallate
**67**	21–23	343 328	44.1	343.04787 C_17_H_11_O_8_ (5.6)	Ellagic acid trimethyl ether
**68**	59–79	349 197	13.9	349.0416 C_16_H_13_O_9_ (42.7) *	Galloyl ethyl gallate
**69**	105–119	425 301	15.8	425.01469 C_20_H_9_O_11_ (0.8)	Ellagic acid pyrogallol ether
**70**	103–119	469 425	15.7	469.0039 C_21_H_9_O_13_ (2.1)	Valoneic acid dilactone
**71**	161–165	481 **439**, 331, 301, 169	1.2	481.06556 C_20_H_17_O_14_ (6.6)	HHDP hexoside
**72**	155–165	483 **439**, 331, 313, 169	11.5	483.07806 C_20_H_19_O_14_ (0.1)	Digalloyl hexoside
**73**	85–89	497 301	23.4	497.03631 C_23_H_13_O_13_ (0.3)	Valoneic acid dilactone ethyl ether
**74**	83–91	625 471, **301**	28.6	625.07458 C_26_H_25_O_18_ (28.1) *	Ellagic acid dihexoside
**75**	147–155	631 **479**, 301	19.4	631.09323 C_27_H_19_O_18_ (26.3) *	NHDP hexoside
**76**	155–165	633 479, **301**	7.8	633.07511 C_27_H_21_O_18_ (2.8)	HHDP galloyl hexoside
**77**	133–151	635 483, **465**, 313	15.6	635.08832 C_27_H_23_O_18_ (1.0)	Trigalloyl hexoside
**78**	135–139	733 635	n.d.	-	Trigalloyl hexoside derivative
**79**	149	781 631, **301**	3.7	781.06132 C_34_H_21_O_22_ (10.7)	Punicalin
**80**	155–167	783 481, **301**	2.7	783.07063 C_34_H_23_O_22_ (2.5)	DiHHDP hexoside
**81**	133–165	785 633, 481, **301**, 275	9.6	785.08378 C_34_H_25_O_22_ (0.7)	HHDP digalloyl hexoside
**82**	133–155	787 635, 617, 483, 465, **301**	21.1	787.09741 C_34_H_27_O_22_ (3.2)	Tetragalloyl hexoside
**83**	145–167	935 917, **633**, 571, 365, 329, 299, 275	6.0	935.07728 C_41_H_27_O_26_ (2.5)	Galloyl diHHDP hexoside
**84**	131–169	937 **785**, 769, 633, 617, 301	6.0	937.28345 C_41_H_29_O_26_ (12.6) *	HHDP trigalloyl hexoside
**85**	155–167	939 **787**, 769, 617, 465	26.2	939.11228 C_41_H_31_O_26_ (1.5)	Pentagalloyl hexoside
**86**	155–169	951 **907**, 783, 605	-	-	DiHHDP hexoside derivative
Condensed tannins					
**87**	121–133	577 **463**, 425, 313, 289	3.9	577.13515 C_30_H_25_O_12_ (2.9)	(Epi)-catechin dimer
**88**	123–125	593 575, 467, **441**, 425, 305	2.6	593.15119 C_30_H_25_O_13_ (4.3)	(Epi)-catechin-(epi)-gallocatechin dimer
**89**	107–119	609 **457**, 439, 321, 169	4.1	609.12858 C_30_H_25_O_14_ (5.9)	(Epi)-gallocatechin dimer
**90**	125–129	897 745, 575, **463**, 449, 423	13.0	897.14880 C_44_H_33_O_21_ (3.6)	(Epi)-catechin gallate -(epi)-gallocatechin gallate dimer
**91**	123–131	913 **463**, 449, 317	8.0	913.14548 C_44_H_33_O_22_ (1.5)	(Epi)-gallocatechin gallate dimer
**92**	137–155	913 **761**, 573, 449, 423	24.6	913.16762 C_45_H_37_O_21_ (17.1)	(Epi)-gallocatechin trimer
Others					
**93**	73–85	109 -	3.0	109.02893 C_6_H_5_O_2_ (5.3)	Catechol
**94**	67–81	124 -	n.d.	-	Amino catechol
**95**	93–115	125 -	12.1	125.02756 C_6_H_5_O_3_ (17.2)	Pyrrogallol
**96**	77–83	153 109	9.1	153.02038 C_7_H_5_O_4_ (6.9)	Protocatechuic acid
**97**	23–25	167 125	9.1	167.03454 C_8_H_7_O_4_ (2.6) *	Vanillic acid
**98**	67–79	168 124	n.d.	-	Amino protocatechuic acid
**99**	85–95	193 111	9.1	193.01665 C_9_H_5_O_5_ (12.5)	Trihydroxychromone
**100**	15	209 187, **165**, 125	n.d.	-	Jasmonic acid
**101**	59–69	217 155	-	-	Unknown
**102**	13–15	279 277, **243**, 237	73.4	279.23401 C_18_H_31_O_2_ (3.8)	Linoleic acid
**103**		281 277, **255**	75.8	281.24987 C_18_H_33_O_2_ (4.5)	Oleic acid
**104**	11–15	295 277, **275**, 265, 251, 249, 185	70.2	295.2304 C_18_H_31_O_3_ (8.6)	Hydroxy linoleic acid
**105**	125–131	305 221, **219**, 179, 165, 125	12.1	305.06942 C_12_H_17_O_7_S (2.1)	5′-hydroxysulphonyloxy jasmonic acid
**106**	11–17	383 **337**	-	-	Unknown
**107**	157–159	707 687, 671, 533, **359**	n.d.	-	Integracin D
**108**	97–99	875 **441**, 433, 289	-	-	Unknown
**109**	89–93	887 **585**, 301	-	-	Unknown

MS^2^ numbers in bold indicate the most intense product ion. * indicate very minor compounds. HHTP = hexahydroxydiphenoyl ester; NHTP = nonahydroxytriphenoyl ester.

## Data Availability

Please, contact the corresponding author for access to database.

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
