# Peer review of "Laguncularia racemosa Phenolics Profiling by Three-Phase Solvent System Step-Gradient Using High-Performance Countercurrent Chromatography with Off-Line Electrospray Mass-Spectrometry Detection"

_molecules, 2021, doi:10.3390/molecules26082284_

Round 1
Reviewer 1 Report
Work submitted for review concerns high-performance countercurrent chromatography (HPCCC) of Laguncularia racemosa extract. Owing to gradient elution in step-gradient elution mode, the authors managed to identify an impressive number of metabolites, which was possible thanks to an off-line injection of fractions into ESI-MS / MS. The whole work is impressive. The authors identified many secondary metabolites and applied appropriate and advanced analytical techniques. There is not much work on this issue, therefore I recommend this work for publication.
However, before publishing, Authors are kindly requested to consider the following comments:
- supplementation of the literature with the following items on CCC in three-phase systems.
Wang, F., Li, R., Long, L. et al. A Three-Phase Solvent System in High-Speed Counter-Current Chromatographic for the Separation and Purification of Bioactive Constituents from Acanthus ilicifolius . Chromatographia 78, 1401–1407 (2015). https://doi.org/10.1007/s10337-015-2952-5
Shinomiya K, Ito Y (2006) J Liq Chromatogr Relat Technol 29:733–750
Shibusawa Y, Yamakawa Y, Noji R, Yanagida A, Shindo H, Ito Y (2006) J Chromatogr A 1133:119–125
Yanagida A, Yamakawa Y, Noji R, Oda A, Shindo H, Ito Y, Shibusawa Y (2007) J Chromatogr A 1151:74–81
Yin H, Zhang S, Long L, Yin H, Tian X, Luo X, Nan H, He S (2013) J Chromatogr A 1315:80–85
Wu X, Chao Z, Wang C, Yu L. Separation of chemical constituents from three plant medicines by counter-current chromatography using a three-phase solvent system at a novel ratio. J Chromatogr A. 2015 Mar 6;1384:107-14. doi: 10.1016/j.chroma.2015.01.057
- Authors used the tri-phasic system n-hexane - tert- butyl methyl ether – acetonitrile – water 2:3:3:2 (v/v/v/v). Please check whether they are recommended as green, pro-ecological solvents. The so-called green solvents must meet a number of criteria, ie low toxicity to humans and other organisms, safe to use, and easily degraded when emitted to the environment. When selecting a solvent for the analytical application, it should be assessed from the point of view of safety of use. Environmentally hazardous reagents should be avoided in analytical laboratories. For this purpose, I recommend reading:
Prat, D., Hayler, J., & Wells, A. (2014). A survey of solvent selection guides. Green Chem., 16 (10), 4546–4551. doi: 10.1039 / c4gc01149j
Author Response
Reviewer 1:
Work submitted for review concerns high-performance countercurrent chromatography (HPCCC) of Laguncularia racemosa extract. Owing to gradient elution in step-gradient elution mode, the authors managed to identify an impressive number of metabolites, which was possible thanks to an off-line injection of fractions into ESI-MS / MS. The whole work is impressive. The authors identified many secondary metabolites and applied appropriate and advanced analytical techniques. There is not much work on this issue, therefore I recommend this work for publication.
R: The authors would like to thank the reviewer for careful reading and suggestions in order to improve the manuscript.
However, before publishing, Authors are kindly requested to consider the following comments:
- supplementation of the literature with the following items on CCC in three-phase systems.
Wang, F., Li, R., Long, L. et al. A Three-Phase Solvent System in High-Speed Counter-Current Chromatographic for the Separation and Purification of Bioactive Constituents from Acanthus ilicifolius . Chromatographia 78, 1401–1407 (2015).
https://doi.org/10.1007/s10337-015-2952-5
Shinomiya K, Ito Y (2006) J Liq Chromatogr Relat Technol 29:733–750
Shibusawa Y, Yamakawa Y, Noji R, Yanagida A, Shindo H, Ito Y (2006) J Chromatogr A 1133:119–125
Yanagida A, Yamakawa Y, Noji R, Oda A, Shindo H, Ito Y, Shibusawa Y (2007) J Chromatogr A 1151:74–81
Yin H, Zhang S, Long L, Yin H, Tian X, Luo X, Nan H, He S (2013) J Chromatogr A 1315:80–85
Wu X, Chao Z, Wang C, Yu L. Separation of chemical constituents from three plant medicines by counter-current chromatography using a three-phase solvent system at a novel ratio. J Chromatogr A. 2015 Mar 6; 1384:107-14. doi: 10.1016/j.chroma.2015.01.057
R: Two references, Chromatographia 78, 1401–1407 (2015) and J Chromatogr A 1384, 107-14 (2015), were added as numbers [53] and [54]. The other references were already on the references list.
- Authors used the tri-phasic system n-hexane - tert- butyl methyl ether – acetonitrile – water 2:3:3:2 (v/v/v/v). Please check whether they are recommended as green, pro-ecological solvents. The so-called green solvents must meet a number of criteria, ie low toxicity to humans and other organisms, safe to use, and easily degraded when emitted to the environment. When selecting a solvent for the analytical application, it should be assessed from the point of view of safety of use. Environmentally hazardous reagents should be avoided in analytical laboratories. For this purpose, I recommend reading:
Prat, D., Hayler, J., & Wells, A. (2014). A survey of solvent selection guides. Green Chem., 16 (10), 4546–4551. doi: 10.1039 / c4gc01149j
R: Since the main aim of this work was to isolate (at semi-preparative scale) and identify as many metabolites as possible, the authors did not take into consideration “greenness" of solvents used. Also, the experimental work (including solvents’ disposal) was done in line with the EU/UK health and safety regulations. Based on the guide suggested by the reviewer, the solvent system spans across “red-orange-yellow-green” range with hexane representing “hazardous” category and water representing “recommended” one. However, the authors will use this advice for future work.
Reviewer 2 Report
The article by Costa et.al. introduces the determination of more than 100 compounds in Laguncularia racemosa extract. Furthermore, it uses four different complex analytical methods to do so. In my opinion, this research contains exceptional work and a thorough knowledge of the field to back it up, which makes it highly desirable to publish. Where it lacks the quality of the latter is the presentation. My main feeling after I read it was confusion. It has a few key issues and several smaller ones, which led to my conclusion. I will give you that in detail.
I. Main issues:
-The confusion of the main theme. In the title, the introduction, and the conclusion we only see the main method (HPCCC-offline ESI-MS/MS). However, from the text, we get that, other methods were also
highly utilized (TLC, HPLC-TOF). The understanding gets tangled when these other methods are introduced.
-I do not see the logic behind choosing what makes the supplementary and what gets into the main body of work. It connects to the previous issue. It would be better if all the steps of the identification were included in the main work. And in reverse, Figure 1. has no real meaning in the process, but takes up almost a whole page.
-We do not get the whole picture of how to connect the dots from start to finish. Often the results are just presented, but not explained. For example, in line 110 we are told the steps in the process, but they are not explained.
-We do not see any mass spectra. This connects to all the issues. The main method to identify the compounds is the tandem mass spectra, but we never see one.
II. Small details that connect to the main issues, and others:
-The introduction is short. The whole first part of the results can be integrated into it. Also, later other information and references get introduced, which can be a part of that too.
-Parts of the introduction are not clear and not detailed enough. For example, the tri-phasic system contains four components. It would be more efficient if they stated clearly the three phases. It gets even more confusing if somebody goes to the methods section because now it has eight components (line 259).
-Figures have no meaningful explanation. They are just presented. It makes them redundant in the context of the whole article. For example, line 94 about Figure 1.
-Line 99. How does it guide us? Who and how does isolated them? Where are they listed?
-Line 103-105. How did it help? Reference?
-Line 108-109. Is this an assumption from previous work or is this part of the result? If the first, where is the reference? If the second (or in any way), how is this connects to the compounds of Table 1? The reader can ultimately connect the dots, but it would be nicer to see that in the article.
-Figure 1. It is not necessary to show us all 17 cases. Include one example, and the rest can go to supplementary.
-Line 129. The references are here, how the MS determination is made. It shows us that the work is behind it, no doubt. However, it is necessary to include it in this article in some form.
-Line 135-140. This list and others like this later are unnecessary and feel like repetition.
-Line 194. Again, the process is told, but not shown.
-Table 1:
-Fractions: There are a lot of compounds present in more fractions and there are several overlaps. The setup was one fraction at every minute (from section 3.5.4). How was that determined? Can that be improved upon?
-ESI/TOF MS Formula: Where does this come from?
-Error in ppm: What is it exactly? What is the range? Are the individual values good or bad?
-References: They are great inclusions. I honestly appreciate the effort. However, because not every compound has one, it begs the question: Are they affect the outcome of this work? What about the ones, that do not have one?
-I have to add, the work behind this table is still impressive.
-Line 223. EtOAcPart is only introduced here and previously used.
-Line 290-291. What about the even-numbered fractions? I do not get this part.
-Line 292-295. Why not direct syringe pump injection is used? This way the sample will be diluted with the eluent without any benefit. Or if the available instrument only has this option, why not include another level of chromatography and add a column? I understand, that it further complicates the methodology, but it will be really interesting to do 2D chromatography like this.
-Line 297-298. Why only the 9 most intense peaks were selected? Did we lose some components with this?
In conclusion: My suggestion is that the authors should include all the methods in the main body of work. Also, a clear and detailed step-by-step description is needed for the determination of the components. Not all of them, because we have more than 100, but maybe one or two examples for every group.
Author Response
The article by Costa et.al. introduces the determination of more than 100 compounds in Laguncularia racemosa extract. Furthermore, it uses four different complex analytical methods to do so. In my opinion, this research contains exceptional work and a thorough knowledge of the field to back it up, which makes it highly desirable to publish. Where it lacks the quality of the latter is the presentation. My main feeling after I read it was confusion. It has a few key issues and several smaller ones, which led to my conclusion. I will give you that in detail.
R: The authors would like to thank the reviewer for careful reading and suggestions in order to improve the manuscript.
Main issues:
-The confusion of the main theme. In the title, the introduction, and the conclusion we only see the main method (HPCCC-offline ESI-MS/MS). However, from the text, we get that other methods were also highly utilized (TLC, HPLC-TOF). The understanding gets tangled when these other methods are introduced.
R: The authors consider TLC and HPLC-Q-TOF-MS as routinely used in labs for quick scanning of samples. TLC is particularly often used in CCC method development as a quick analytical method for solvent system searches and separation tracking and visualization method of separated compounds, especially when dealing with non-UV active molecules. HPLC-Q-TOF-MS is the tool to generate sum formula of unknown metabolites, also quite common in natural product analysis. Whereas the off-line -ESI-MS/MS injection profiling methodology is one of the sophisticated techniques used in this HPCCC study on tri-phasic solvent systems to monitor the tremendous diversity of separated metabolites, and was therefore described and discussed in more detail.
-I do not see the logic behind choosing what makes the supplementary and what gets into the main body of work. It connects to the previous issue. It would be better if all the steps of the identification were included in the main work. And in reverse, Figure 1. has no real meaning in the process, but takes up almost a whole page.
R: Thank you for the suggestion. Figures from the manuscript and supplementary material part were reviewed and explanation in section 2.1 were added to make the approach clearer. However, Figure 1 was not removed to the Supplements as it is a central part of the HPCCC investigation. It is the most fast metabolite profiling tool to screen preparative fractions for interesting molecular entities. The selected ion traces of the 17 principal metabolites clearly indicate the chromatographic performance of the tri-phasic HPCCC separation approach. The visualization of separated compounds, and co-elution effects is an important result.
We added some more explanations to make the section of the HPCCC and off-line ESI-MS with the central Figure 1 more meaningful.
-We do not get the whole picture of how to connect the dots from start to finish. Often the results are just presented, but not explained. For example, in line 110 we are told the steps in the process, but they are not explained.
R: Some more explanations were added in the second paragraph of Section 2.1.
-We do not see any mass spectra. This connects to all the issues. The main method to identify the compounds is the tandem mass spectra, but we never see one.
R: The authors were concerned to overload the manuscript and the supplementary material with raw data for more than 100 compounds. Therefore, the decision was made to provide a summary of the extracted ESI-MS data as a table (MS and MS/MS column).
Small details that connect to the main issues, and others:
-The introduction is short. The whole first part of the results can be integrated into it. Also, later other information and references get introduced, which can be a part of that too.
R: The work presented in this manuscript is original and novel and therefore, there is not much background information found in the literature to be used in Introduction section. The authors feel that the current layout and the style of the manuscript allows a better flow of the story and helps with discussion of the results.
-Parts of the introduction are not clear and not detailed enough. For example, the tri-phasic system contains four components. It would be more efficient if they stated clearly the three phases. It gets even more confusing if somebody goes to the methods section because now it has eight components (line 259).
R: While describing solvent systems the authors used a well-established terminology in CCC separation technology, which was originally adopted from liquid-liquid extraction. A biphasic or triphasic solvent system, commonly used in CCC separations, could contain from 2 to 5 solvents/components (hexane-water is an example). The number of solvents does not reflect the number of layers. For example, hexane-methanol-water mixtures (at certain ratios) are three component systems but form two layers. A triphasic system can be composed of 4 solvents.
In the methods section (line 279), two solvent systems containing four solvents each were tested (n-hexane – methyl acetate –acetonitrile – water and n-hexane – tert-butyl methyl ether – acetonitrile – water). This is a standard procedure in the CCC field.
Line 259 (in the original manuscript) is the description of LC-ESI/TOF MS preliminary analysis.
-Figures have no meaningful explanation. They are just presented. It makes them redundant in the context of the whole article. For example, line 94 about Figure 1.
R: Thank you for the comment. Figures from the manuscript and supplementary material were reviewed and re-organized. The corresponding text and captions were also revised, and additional explanations were included where necessary
-Line 99. How does it guide us? Who and how does isolated them? Where are they listed?
R: The authors used literature data for previously isolated compounds. This part has been re-written to make it clear. Please, verify the second paragraph of Section 2.1.
-Line 103-105. How did it help? Reference?
R: Lines 103-105: “High accuracy molecular weights acquired by LC-ESI/TOF MS were used to ratify and/or verify the proposed molecular formulas. Chemotaxonomic knowledge of previously identified compounds in other genus of Combretaceae helped to support the results.”
High accuracy molecular weights were used to compare and confirm proposed structures. Also, the proposed structures were compared to previously isolated compounds in Combretaceae to check if the family have the capacity to biosynthesize such compounds. References were added alongside the text, for example, in the identification of flavonoids: “Aglyca apigenin, kaempferol, quercetin and tricin were previously reported in L. racemosa [8, 26] in addition to the glycosylated derivatives quercetin-3-O-arabinoside and quercetin-3-O-rhamnoside [24].”
-Line 108-109. Is this an assumption from previous work or is this part of the result? If the first, where is the reference? If the second (or in any way), how is this connects to the compounds of Table 1? The reader can ultimately connect the dots, but it would be nicer to see that in the article.
R: The authors used literature data for previously isolated compounds. This part has been re-written to make it clearer. Please, verify the second paragraph of Section 2.1.
-Figure 1. It is not necessary to show us all 17 cases. Include one example, and the rest can go to supplementary.
R: Thank you for the suggestion. Figures in the manuscript and supplementary material were reviewed and re-organized.
-Line 129. The references are here, how the MS determination is made. It shows us that the work is behind it, no doubt. However, it is necessary to include it in this article in some form.
R: The authors felt that it would be better not to show all fragmentations one by one because those compounds are known and very common. Also, the fragmentation pattern of phenolics is well established and can be easily found in the cited references.
-Line 135-140. This list and others like this later are unnecessary and feel like repetition.
-Line 194. Again, the process is told, but not shown.
R: This part of the manuscript is describing the MS determination process in a succinct way, including fragmentation losses. The fragmentation pattern of phenolics is well established and can be easily found in the cited references.
Table 1:
-Fractions: There are a lot of compounds present in more fractions and there are several overlaps. The setup was one fraction at every minute (from section 3.5.4). How was that determined? Can that be improved upon?
R: The separation methodology was purposely based on a standard CCC protocol for a gradient elution (collecting fractions of 4 mL each from a 143.5 mL column) which generally provides a good separation. The following CCC-MS/MS analysis showed the complexity of EtOAcPart with many compounds present in some fractions. Being able to detect and annotate most of them shows the success of our methodology.
-ESI/TOF MS Formula: Where does this come from?
R: ESI/TOF MS comes from the item 3.4 (LC-ESI/TOF MS preliminary analysis). The formula is a chemical formula of the annotated compound.
-Error in ppm: What is it exactly? What is the range? Are the individual values good or bad?
R: Error was calculated by the software according to the formula (see above). Errors below 10 ppm indicate a good chance to be the proposed formula. Minor compounds could have errors above the good value due to their low concentrations and compound co-elutions. It is difficult to predict how a compound’s concentration might affect the MS and MS/MS analysis.
-References: They are great inclusions. I honestly appreciate the effort. However, because not every compound has one, it begs the question: Are they affect the outcome of this work? What about the ones, that do not have one?
R: The fragmentation references were not available for all isolated compounds which led to mentioning in the text but not in the table.
-I have to add, the work behind this table is still impressive.
R: Thank you!
-Line 223. EtOAcPart is only introduced here and previously used.
R: Thank you for pointing it out. This has been corrected.
-Line 290-291. What about the even-numbered fractions? I do not get this part.
R: Based on their extensive experience in CCC separations, the authors used a standard approach in CCC separation to analyze either odd-numbered or even-numbered fractions in order to reduce time and cost. If two odd-numbered CCC fractions have the same chemical profile, it means that the even-numbered fraction in-between them have identical chemical composition.
-Line 292-295. Why not direct syringe pump injection is used? This way the sample will be diluted with the eluent without any benefit. Or if the available instrument only has this option, why not include another level of chromatography and add a column? I understand, that it further complicates the methodology, but it will be really interesting to do 2D chromatography like this.
R: The off-line injection profiling approach is the experiment to project a preparative CCC-experiment (gram –range) by selected ion traces. That means that the preparative elution order, and of course co-elution effects of metabolites can be visualized and used for the very accurate fractionation process. So there is no unintentional mixing of recovered fractions, as they are visual by their chemical identities (cf. Figure 1). This automated experiment is recorded in a single data file, so detecting the elution of a pure target molecule e.g. in a sequence of fraction tube 5-12, these tubes could be taken out, pooled and probably used for direct NMR analysis. This approach is done in a 60-70 min experiment.
Doing manual syringe pump infusions on this large set of recovered fraction will generate many independent data files which all need to be inspected step-by-step, very time consuming and laborious work (imagine changing 100 times or more the sample fillings of the syringe by hand) and then evaluation and combination of the experimental outcomes in all data files.
Of course, the generation of 2D-plots consisting of the preparative CCC separation combined with LC-ESI-MS of recovered fractions would enable much lower metabolite detection limits but require high experimental times, and would deliver ‘too’ much informations which lately hamper the fastness of decisions which will be required for adequate fractionation of the collected glas vials/tubes. The time factor in this case is critical, as certain natural products are not stable in solvent mixtures for a long time.
This methodology of metabolite profiling for fractionation of complex natural product extracts on preparative scale is well established, and recognized (e.g. Costa, F.N.; Borges, R.M.; Leitão, G.G.; Jerz, G. Preparative mass‐spectrometry profiling of minor concentrated metabolites in Salicornia gaudichaudiana Moq by high‐speed countercurrent chromatography and off‐line electrospray mass‐spectrometry injection. J Sep Sci 2019, 42, 1528-1541.)
-Line 297-298. Why only the 9 most intense peaks were selected? Did we lose some components with this?
R: The authors used the nine most intense peaks approach based on their experience that it is a “safe” number of peaks not to lose any important compound.
In conclusion: My suggestion is that the authors should include all the methods in the main body of work. Also, a clear and detailed step-by-step description is needed for the determination of the components. Not all of them, because we have more than 100, but maybe one or two examples for every group.
R: As mentioned above, TLC and HPLC-TOF are considered to be a routine analysis these days, although the authors understand that it depends on equipment availability. The main aim of the manuscript was to describe a combination of three-phase gradient CCC & ESI-MS/MS for a particular application. The authors felt that the provided discussion is enough for readers to get an understanding of the concept and make them interested in collaboration.
English was corrected and improved by an native speaker (author Peter Hewitson).